# CBM-ZERO: CONCEPT BOTTLENECK MODEL WITH ZERO PERFORMANCE LOSS

## ABSTRACT

Interpreting machine learning models with high-level, human-understandable *concepts* has gained increasing interest. The concept bottleneck model (CBM) is a popular approach to providing interpretable models, relying on first predicting the presence of concepts in a given input, and then using these concept scores to predict a label of interest. Yet, CBMs suffer from lower accuracy compared with standard black-box models, as they use a surrogate (and thus, interpretable) predictor in lieu of the original model. In this work, we propose an approach that allows us to find a CBM in any standard black-box model via an invertible mapping from its latent space to an interpretable concept space. This method preserves the original black-box model's prediction and thus has zero performance drop while providing human-understandable explanations. We evaluate the accuracy and interpretability of our method across various benchmarks, demonstrating state-of-the-art explainability metrics while enjoying superior accuracy.

## 1 INTRODUCTION

As artificial intelligence (AI) demonstrates remarkable success in diverse domains, concerns regarding interpretability (Holzinger et al. (2022); Dwivedi et al. (2023)), fairness (Chen et al. (2023); Bharti et al. (2024)), and privacy (Oseni et al. (2021); Xu et al. (2024)) are also gaining increasing attention. Particularly, while the complexity of deep learning models enables them to model complex patterns, it also makes the decision-making process opaque. This "black-box" nature raises concerns about deploying these models in high-stakes scenarios, and there is a growing demand for more transparent AI systems (Von Eschenbach (2021); Gryz & Rojszczak (2021)).

Numerous efforts have been made to enhance the interpretability of deep learning models. Many of these works generate salience-map style explanations using gradient-based analysis (Simonyan et al. (2013); Selvaraju et al. (2017)), game-theory approaches (Sundararajan & Najmi (2020); Teneggi et al. (2022)), decomposition-based methods (Zhou et al. (2016); Shrikumar et al. (2017)), and more. The salience map highlights important parts of the input (i.e., pixels in images; nodes in graphs; words in sentences, etc.), which provides valuable insights into *where*, or to what features, the model is looking at. However, salience maps might not always be sufficient, specifically when the prediction is based on global attributes, such as color, texture, or overall morphology, rather than specific input dimensions (Poeta et al. (2023)). This limitation is particularly evident in challenging tasks like clinical diagnosis, where localizing a particular subregion in medical images may not fully convey the reasoning of a model, and high-level, domain-relevant explanations are needed for more meaningful explanations (Border & Sarder (2022); Venkatesh et al. (2024)).

Concept-based explanations provide a compelling alternative, which explains classification models by high-level, human-understandable attributes such as color, shape, texture, and the presence of objects (Poeta et al. (2023)). The *concept bottleneck model* (CBM) (Losch et al. (2019); Koh et al. (2020)) is one such method that consists of two interconnected predictors: a first concept predictor that predicts the presence of specific abstract concepts in some embedded representation of the input, and a subsequent (linear) model that outputs the probability of the class given the presence of such concepts. Since this model explicitly constructs predictions that rely on the presence of concepts, this layer is referred to as a *concept bottleneck*. More recently, Contrastive Language-Image Pre-Training (CLIP) models are employed(Radford et al. (2021)) to replace the concept predictor and build the concept bottlenecks by calculating the alignment between certain images and textual

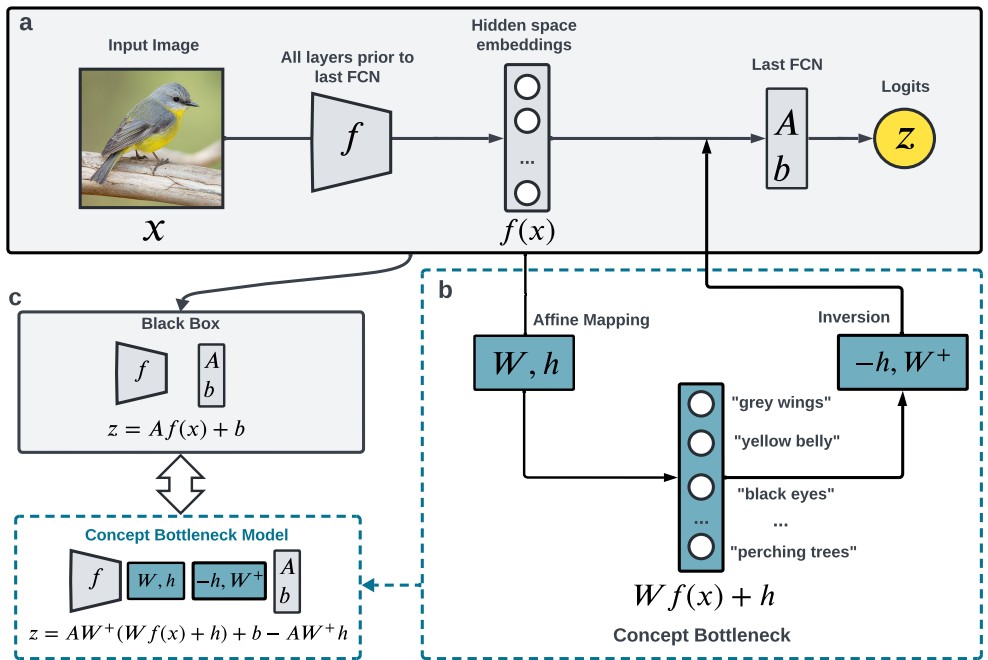

Figure 1: The high-level idea of CBM-zero **(a)** A black-box model. **(b)** Construction of a concept bottleneck through invertible affine mapping from black-box model's hidden space to concept space. **(c)** The black-box model can be reformulated as a CBM without altering predictions.

concepts (Yuksekgonul et al. (2022); Yang et al. (2023); Oikarinen et al. (2023)), alleviating the demand for image-wise concept annotations. A crucial challenge of these CBM-based methods is the loss of predictive power, as they use a surrogate model in lieu of a standard black-box model to gain interpretability. Although numerous efforts have been made, such as constructing very complex and large concept banks Yang et al. (2023); Shang et al. (2024); Schrodi et al. (2024); Hu et al. (2024); Tan et al. (2024), the performance drop still exists, especially in complex tasks.

In this work, we propose CBM-zero, which explains an established, standard black-box model by converting it to a concept bottleneck model that inherently has zero performance drop. As Figure 1 illustrates, CBM-zero extracts the latent space of a black-box model just before its *final* layer and finds an affine mapping from this latent space to a concept space derived from human annotations or CLIP models (which eliminates the need for dense concept annotations). Importantly, we impose an *invertibility* constraint on the affine map to ensure the original black-box model's prediction can be recovered completely. In this way, the original black-box model is converted to a CBM without perturbing its prediction function (see Figure 1.c). Thus, we can explain that black-box model in a post-hoc manner without altering the original predictor, and retaining its performance. We evaluate our method on several image classification benchmarks, including general and fine-grained tasks. Unlike previous CBM-based works, which only show qualitative analysis or partial quantitative results concerning the quality of interpretation, we quantitatively evaluate them by proposing precise metrics with respect to human annotations (when possible), or to large-scale knowledge graphs. Compared with other CBM-based works, our method consistently achieves the best accuracy and offers comparable or better interpretations.

## 2 RELATED WORKS

As alluded to above, our work is most closely related to CBM-based methods. More broadly, it is also related to post-hoc conceptual explanation techniques, given our method aims to explain without altering the original model. We review these prior works in both categories to put our contribution in context.

**Concept bottleneck model (CBM)** The initial idea of CBM (Koh et al. (2020); Losch et al. (2019)) relied on predicting concepts first and then using these predicted concept scores to make a final classification, forming a concept bottleneck. Dense image-wise concept annotations are needed to learn this concept bottleneck. Post-hoc CBM (PCBM) (Yuksekgonul et al. (2022)) proposed learning concept activation vector (CAVs) in feature space as a concept bank (Kim et al. (2018)) and project the image embeddings (extracted by a pre-trained image feature encoder) to CAVs to form the concept bottleneck. Moreover, in cases where image-wise concept annotations are not available, they show that the concept bottleneck can be obtained by aligning images with textual descriptions of concepts through language-vision models such as CLIP (Radford et al. (2021)). Language in the Bottle (LaBo) (Yang et al. (2023)) and label-free CBM (Oikarinen et al. (2023)) follow a similar idea and further boost the accuracy by collecting concepts from a large language model and conducting the concept selection carefully. More recently, increasingly sophisticated methods have been dedicated to improving the concept bank quality from the aspect of completeness (Shang et al. (2024)) and flexibility Schrodi et al. (2024); Hu et al. (2024); Tan et al. (2024). All of these CBM methods have the crucial limitation of performance drops compared to black-box models. While numerous efforts have been made to minimize this performance gap, some prediction power is always lost in all of these approaches. Our method, on the other hand, recovers the black-box model's original prediction exactly and therefore has zero performance drop by design.

**Post-hoc concept explanations** Post-hoc concept explanations typically need image-wise concept annotations in the training set, or an auxiliary set equipped with such annotations. T-CAV (Kim et al. (2018)) is a popular approach that learns a linear classifier in a black-box model's feature space to distinguish samples with and without certain concepts, parameterized by concept activation vectors (CAVs). The importance of the concept is then given by directional derivatives of prediction to CAVs. There are a bunch of extensions that generalize and enhance T-CAV. For instance, Automatic Concept-based Explanations (ACE) (Ghorbani et al. (2019)) consider superpixels in images as concepts and discover them automatically. ConceptSHAP (Yeh et al. (2020)) defines completeness scores for CAVs, and uses Shapley values to quantify the individual importance. Concept activation region (CAR) (Crabbé & van der Schaar (2022)) relaxes the linear separability assumption of CAV and uses a region instead of a vector in latent space to represent a concept. Spatial CAV (Wang & Lee (2022)) attributes CAVs to relevant spatial regions. Text2Concept (Moayeri et al. (2023)) derives CAV from texts. Casual Concept Effect (CaCE) (Goyal et al. (2019)), on the other hand, assesses the causal effect of the presence of concepts by generating counterfactual samples on samples annotated with concepts. More recently, Teneggi & Sulam (2024) uses conditional independence of concepts and sequential kernelized testing (Shekhar & Ramdas (2023)) to assess concept importance in predictions. The vast majority of these methods need costly concept annotations to define concepts in a black-box model's hidden space.

## 3 METHODS

In this section, we outline the problem formulation, describe our methods, and briefly introduce the related methods to be used in the experimental section.

### 3.1 PROBLEM FORMULATION

Consider a deep learning, black-box model that predicts a label[1] $y \in \mathbb{R}$ from an input image $x \in \mathbb{R}^n$. Most deep neural networks consist of multiple stacked layers and end with a fully connected layer (FCN). Let $f : \mathbb{R}^n \to \mathbb{R}^d$ represent all layers prior to the last FCN. Without loss of generality, we consider a $K$-class image classification task. The weights and bias of the final FCN are denoted as $A \in \mathbb{R}^{K \times d}$ and $b \in \mathbb{R}^K$, respectively. The black-box model, and it's prediction $\hat{y}$, are formulated as:

$$z = Af(x) + b, \quad \hat{y} = \arg\max_i z_i, \qquad (1)$$

where $z \in \mathbb{R}^K$ denotes the logits for $K$ classes, and the predicted label $\hat{y}$ is the index of largest logit. The goal is to explain this black-box model with high-level, human-understandable *concepts*

---

[1]Even though we will study classification problems, we will consider labels in $\mathbb{R}$ as we model the unnormalized logits of the model, which approximate the conditional probability of a label given the input.

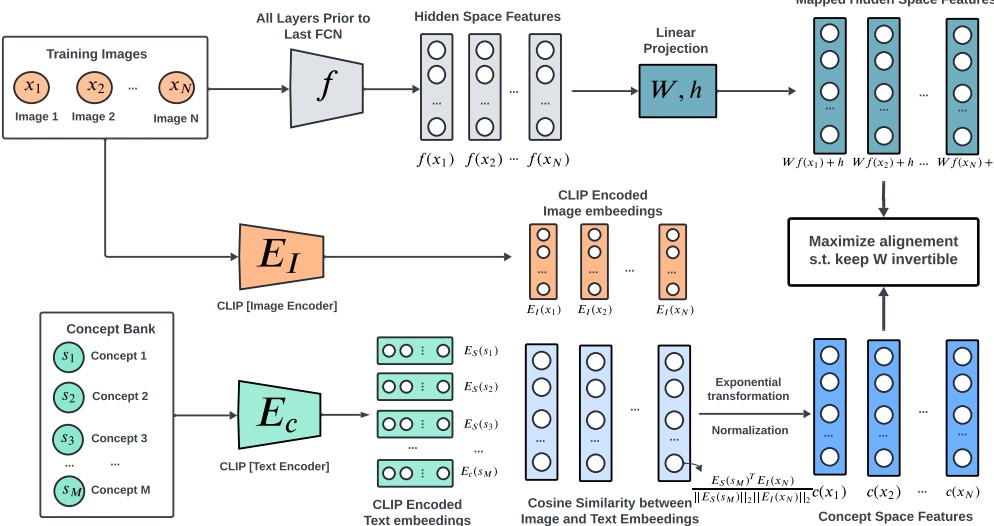

Figure 2: **An overview of the method.** CLIP model is used to encode images and textual concepts. CLIP scores (i.e., cosine similarity between image embeddings and text embeddings) provide an estimation of certain concepts' correlation with certain images. Exponential transformation and normalization are applied to emphasize highly correlated concepts, yielding concept space features for images. An invertible linear mapping, $(W, h)$ is learned to map image features from a black-box model's hidden space to the concept space.

(e.g., "red", "beak", "stripes"). Like other CBM-based methods, we attempt to solve this problem by defining a *concept bank* as a collection of $M$ relevant concepts, denoted as $S = \{s_1, s_2, ..., s_M\}$. We seek to construct a CBM by mapping $f(x)$ to a concept space where these $M$ concepts reside and then inverse the mapping to recover the original prediction exactly. We now move to present details of the proposed approach.

## 3.2 CONSTRUCTING ZERO-PERFORMANCE DROP CBM

In this section, we will show that any black-box model with the form of Equation 1 can be converted to an interpretable CBM without changing the predictions and thus preserving the accuracy of the original model. Figure 2 shows an illustration of our method. The core challenge with black-box models is the lack of interpretability in their latent space. For example, the feature embedding $f(x)$ in Equation 1 resides in an abstract space that encodes useful information for prediction but is not necessarily semantically meaningful or understandable by humans. Thus, we aim to relate this uninterpretable hidden space with another space that is interpretable to humans by construction.

In some cases, images are annotated with concept labels present in the image (in addition to their class label), such as the CUB-200-2011 dataset (Wah et al. (2011)). However such dense concept annotations are not always available. Therefore, we use the CLIP model (Radford et al. (2021)) to estimate the presence of concepts in images in a way that no images-wise concept annotations are needed for training. CLIP trains an image encoder $E_I : \mathbb{R}^n \to \mathbb{R}^l$ and a text encoder $E_S : \mathcal{S} \to \mathbb{R}^l$ (where $\mathcal{S}$ is the space of text/token sequences) jointly via contrastive learning, allowing image and text embeddings to live in a shared space. Their cosine similarity is defined as the CLIP score:

$$\cos(x, s_i) = \frac{E_I(x)^T E_S(s_i)}{\|E_I(x)\|_2 \|E_S(s_i)\|_2}. \tag{2}$$

For an image $x$, the CLIP scores for $M$ concepts yields a vector, $[\cos(x, s_1), \cos(x, s_2), \ldots, \cos(x, s_M)]^T$. To address the limited discriminative power of CLIP (Chattopadhyay et al. (2024)), we apply an exponential transformation to emphasize concepts with stronger correlations, followed by normalization. Specifically, the concept features $c(x)$ is defined as $c(x) = \left[\frac{\cos^t(x, s_1) - \mu_1}{\sigma_1}, \ldots, \frac{\cos^t(x, s_M) - \mu_M}{\sigma_M}\right]^T$, where $\mu_i$ and $\sigma_i$ are the mean and standard

---

**Algorithm 1** Make Matrix Full Rank

---

1: **Input:** A rank-deficient matrix $W \in \mathbb{R}^{M \times d}(M \geq d)$, tolerance scale $\epsilon$
2: **Output:** A full-rank matrix $W'$
3: Perform singular value decomposition (SVD) of $W$: $W = U\Sigma V^T$
4: **for** each singular value $\sigma_i$ in $\Sigma$ **do**
5:    **if** $\sigma_i = 0$ **then**
6:       Sample $r \sim U(0,1)$
7:       Set $\sigma_i' = \epsilon r$ (small perturbation)
8:    **else**
9:       Set $\sigma_i' = \sigma$ (keep original singular value)
10:   **end if**
11: **end for**
12: Set $\Sigma' = \text{diag}(\sigma_1', \dots, \sigma_d')$
13: Reconstruct the matrix $W' = U\Sigma'V^T$
14: **Return** $W'$

---

deviation of $cos^t(x, s_i)$ over the input space $\mathcal{X}$, estimated from training samples. We set $t > 1$ to emphasize concepts with higher CLIP scores (see details in Section 4).

We define an interpretable concept space $\mathcal{C} \subset \mathbb{R}^M$ where $c(x)$ resides, and a hidden space $\mathcal{H} \subset \mathbb{R}^d$ where $f(x)$ resides. We seek an affine projection, $(W \in \mathbb{R}^{M \times d}, h \in \mathbb{R}^M) : \mathcal{H} \to \mathcal{C}$ to map from the hidden space to the concept space. Importantly, we impose the constraint that $\text{rank}(W) = d$, and therefore $M \geq d$, to ensure that the left pseudo-inverse of $W$, defined by $W^+ = (W^T W)^{-1} W^T$, exists. This ensures that the mapping from hidden space to concept space is invertible, allowing us to preserve the output of the original black-box model unaltered.

To learn $W$ and $h$, we solve the following optimization problem under the rank constraint:

$$(W, h) = \arg\min_{W,h} \mathbb{E}_{x \sim \mathcal{D}} \|Wf(x) + h - c(x)\|_2^2 + \lambda R(W) \quad \text{s.t.} \quad \text{rank}(W) = d \quad (3)$$

where $\mathcal{D}$ is the training data distribution, $R(W)$ is a regularization term, and $\lambda$ controls regularization strength. We use elastic net regularization on $W^+$ to encourage its sparsity, which facilitates interpretability. To be more specific,

$$R(W) = \alpha\|W^+\|_1 + (1 - \alpha)\|W^+\|_F^2, \quad (4)$$

where $\|\cdot\|_F$ is the Frobenius norm, $\|W^+\|_1 = \sum_i \sum_j |(W^+)_{i,j}|$ is element-wise $\ell_1$ norm, and $\alpha$ controls this trade-off.

Practically, for each black-box model, we train a single-layer linear model to learn $W$ and $h$ with Adam optimizer (Kingma & Ba (2017)) on the training set (where the black-box model is trained), via the objective in Equation 3. The concepts present in $c(x)$ come from a concept bank, $S$, which is a task-specific set containing concepts relevant to the prediction task of interest (we describe this further in Section 4). The rank of $W$ is tracked each time the linear model is updated. When the full-rank constraint is not fulfilled, we add some small perturbation to its zero singular values, as detailed in Algorithm 1.

### 3.3 GLOBAL AND LOCAL EXPLANATIONS

After $W$ and $h$ are learned, the original black-box model's logits can be reformulated as a linear combination of $M$ interpretable concepts without modifying the original predictor:

$$z = \tilde{A}(Wf(x) + h) + b - \tilde{A}h = Af(x) + b, \quad (5)$$

where $Wf(x) + h \in \mathbb{R}^M$ represents the features in the concept space, $\tilde{A} := AW^+$ provides the adapted linear classifier, and $b - \tilde{A}h$ is the sample-independent (corrected) bias. Notice that, since $W^+W = I$, these logits have not changed from those in the original prediction. Yet, this expression allows us to compute the difference in importance of a given concept to a specific class. To understand the prediction of class $i$, we can compute the deviation of $z_i$ from the mean logit across classes:

$$z_i - \frac{1}{K}\sum_{j=1}^{K} z_j = \sum_{m=1}^{M} \underbrace{(\tilde{A}_{i,m} - \frac{1}{K}\sum_{j=1}^{K}\tilde{A}_{j,m})}_{\Gamma_{i,m}}(Wf(x)+h)_m + B, \tag{6}$$

where $B = b_i - \frac{1}{K}\sum_j b_j + \frac{1}{K}\sum_j (\tilde{A}h)_j - (\tilde{A}h)_i$ is a constant term. From this, we can identify the *global* importance of concept $s_m$ in predicting class $i$, $\Gamma_{i,m} \coloneqq \tilde{A}_{i,m} - \frac{1}{K}\sum_j \tilde{A}_{j,m}$. Moreover, $\gamma_{i,m}(x) \coloneqq \Gamma_{i,m}(Wf(x)+h)_m$ is the *local* contribution of concept $s_m$ to the specific sample $x$. For those particularly interested in how the model discriminates class $i$ from class $j$, the difference between these two classes' logits can be calculated by:

$$z_i - z_j = \sum_{m=1}^{M}(\tilde{A}_{i,m} - \tilde{A}_{j,m})(Wf(x)+h)_m + b_i - b_j + (\tilde{A}h)_j - (\tilde{A}h)_i. \tag{7}$$

These definitions will become useful later in Section 4.

### 3.4 COMPARATIVE METHODS

We compare our method with several CBM-based approaches, all of which employ CMBs coupled with vision-language models like CLIP, but differ in their strategy for concept collection and in how they predict labels.

**Post-hoc CBM (PCBM) (Yuksekgonul et al. (2022))** This method has two versions depending on whether dense concept annotations are available. In cases with annotations, PCBM trains a linear classifier to learn concept activation vectors (CAV) (Kim et al. (2018)), and projects the image embedding onto these CAVs to obtain concept scores. In cases without annotations, CLIP is used to encode the image and textual concepts, and the CLIP-derived image embedding vectors are projected onto text embeddings to get concept scores. In both versions a sparse linear classifier is trained on these concept scores to make final predictions. This method can also be extended by a hybrid PCBM (PCBM-h), which introduces an uninterpretable residual predictor to boost prediction accuracy, but compromises interpretability.

**Language in a Bottle (LaBo) (Yang et al. (2023))** This method assumes ground-truth concept annotations are not available. The scheme is similar to the CLIP version of PCBM, but the concept score is defined as the inner product of image embedding and concept embedding, instead of the projection length. A linear predictor with softmax-normalized coefficients is then trained to predict labels, with no sparsity regularization.

**Label free CBM (LF-CBM) (Oikarinen et al. (2023))** LF-CBM first learns a linear mapping from a black-box model's hidden space to a concept space, and then trains a sparse linear predictor on the mapped image features to predict labels. While the mapping from hidden space to concept space shares a similar motivation to our method, it learns a new predictor to predict labels – thus, it cannot explain the original prediction, and typically results in a loss of predictive power. In contrast, our method learns an invertible mapping, ensuring the original model is preserved.

These methods also differ in the ways that their respective concept banks are constructed: PCBM queries ConceptNet (Speer et al. (2017)), while LaBo and LF-CBM query large language models. Our method is flexible, and we consider various concept sources for different types of datasets, as expanded in Section 4.

## 4 EXPERIMENTS

Herein we first present the experiment setup and evaluation metrics, and then numerical results with comparisons with prior works.

**Datasets and Concept banks** We evaluate our method on six datasets in total: three for standard image classification sets (CIFAR-10, CIFAR-100 (Krizhevsky et al. (2009)) and ImageNet-1K (Russakovsky et al. (2015))), and three fine-grained image classification datasets (CUB-200-2011

Table 1: Accuracy and X-Factuality of Global Explanation on General Image Classification sets

| Method | CIFAR-10 | | CIFAR-100 | | ImageNet-1K | |
|---|---|---|---|---|---|---|
| | ACC | X-Fact@10 | ACC | X-Fact@10 | ACC | X-Fact@10 |
| Black- box | 0.979 | – | 0.873 | – | 0.844 | – |
| PCBM | 0.937 | $0.650 \pm 0.136$ | 0.826 | $\mathbf{0.501 \pm 0.173}$ | 0.814 | $0.307 \pm 0.158$ |
| LaBo | 0.976 | $0.540 \pm 0.080$ | 0.855 | $0.422 \pm 0.159$ | 0.830 | $0.225 \pm 0.119$ |
| LF-CBM | 0.978 | $0.670 \pm 0.142$ | 0.848 | $0.375 \pm 0.131$ | 0.708 | $0.248 \pm 0.126$ |
| CBM-zero(Ours) | $\mathbf{0.979}$ | $\mathbf{0.720 \pm 0.125}$ | $\mathbf{0.873}$ | $0.456 \pm 0.163$ | $\mathbf{0.844}$ | $\mathbf{0.316 \pm 0.163}$ |

Table 2: Accuracy and X-Factuality of Global Explanation on Fine-grained Image Classification sets

| Method | CUB-200-2011 | | AwA2 | | Food-101 | |
|---|---|---|---|---|---|---|
| | ACC | X-Fact@10 | ACC | X-Fact@10 | ACC | X-Fact@10 |
| Black- box | 0.861 | – | 0.981 | – | | – |
| PCBM | 0.824 | $0.119 \pm 0.123$ | 0.632 | $0.486 \pm 0.201$ | 0.947 | $0.413 \pm 0.185$ |
| LaBo | – | – | 0.897 | $\mathbf{0.674 \pm 0.161}$ | 0.943 | $0.335 \pm 0.161$ |
| LF-CBM | 0.831 | $0.268 \pm 0.171$ | 0.976 | $0.673 \pm 0.136$ | 0.944 | $0.545 \pm 0.171$ |
| CBM-zero(Ours) | $\mathbf{0.861}$ | $\mathbf{0.598 \pm 0.120}$ | $\mathbf{0.981}$ | $0.616 \pm 0.146$ | $\mathbf{0.953}$ | $\mathbf{0.556 \pm 0.187}$ |

(Wah et al. (2011)), AwA2 (Xian et al. (2018)), and Food-101 (Bossard et al. (2014))). Each dataset has an associated concept bank relevant to its task. We use existing concept annotations for CUB-200-2011 and AwA2 (bird and animal classification, resp.). CUB-200-2011 contains image-wise annotations for 312 concepts describing detailed visual features of a bird, such as "*has wing color: blue*". AwA2 has class-wise annotations for 85 concepts describing color and characteristic parts in animals, such as *"black", "hairless", "hooves"*. For CIFAR-10, CIFAR-100, and ImageNet-1K, we curate 85, 691, and 2,901 concepts, respectively, by querying ConceptNet (Speer et al. (2017)) with class names (see details in Appendix A.4). For Food-100, we use 1,295 concepts curated by LaBo (Yang et al. (2023)) using GPT-3 (see details in Appendix A.4), as class names are specific food names and less present in ConceptNet. All methods use the same concept bank per dataset.

**Black-box models and CLIP models** We train a black-box (i.e. general) model for each dataset. Our method works for any black-box model provided that the number of concepts ($M$) is larger than the dimension ($d$) of its last FCN. We used the image encoder of CLIP-ViT-L/14 from OpenAI (Radford et al. (2021)) as the backbone and attach a two-layer multi-layer perceptron (MLP) to it. The hidden dimension of MLP is set to be 64 for CIFAR-10 and AwA2, and 256 for CIFAR-100, CUB-200, and Food-101. For ImageNet-1K, a linear layer replaces the MLP since the number of concepts is very large. During the training of black-box models, the image encoder is fixed across all datasets. Additionally, we use CLIP-ViT-L/14 as the CLIP model for interpretation, employing its image encoder as $E_I$ and text encoder as $E_S$. Results of other versions of CLIP models are also included in Appendix A.5, specifically in Figure A.4.

**Hyperparameters** We empirically set $t = 5$, $\lambda = \frac{2}{d}$ ($d$ is the dimension of $f(x)$), and $\alpha = 0.99$ across all the datasets. We discuss other choices for these parameters and their sensitivity in Appendix A.5, specifically in Figure A.2 and A.3.

**Evaluation metrics** The prediction power of models is easily evaluated by their classification accuracy. The objective evaluation of interpretability, on the other hand, is more challenging. Prior works have often omitted quantitative evaluation (Yuksekgonul et al. (2022)), or relied on subjective human inputs (Yang et al. (2023); Oikarinen et al. (2023)). In this work, we provide quantitative and objective evaluations for both global and local concept attributions, as we detail next.

GLOBAL CONCEPT IMPORTANCE We define a precise metric, termed *X-factuality@k*, to evaluate the validity of k-top concepts receiving the highest global importance scores to explain a given class. As detailed in Section 3.3 and Equation 6, the *global* importance score of concept $s_m$ for class $i$ is defined as $\Gamma_{i,m}$. We rank the $M$ concepts by their importance score, select the top $k$ concepts, and

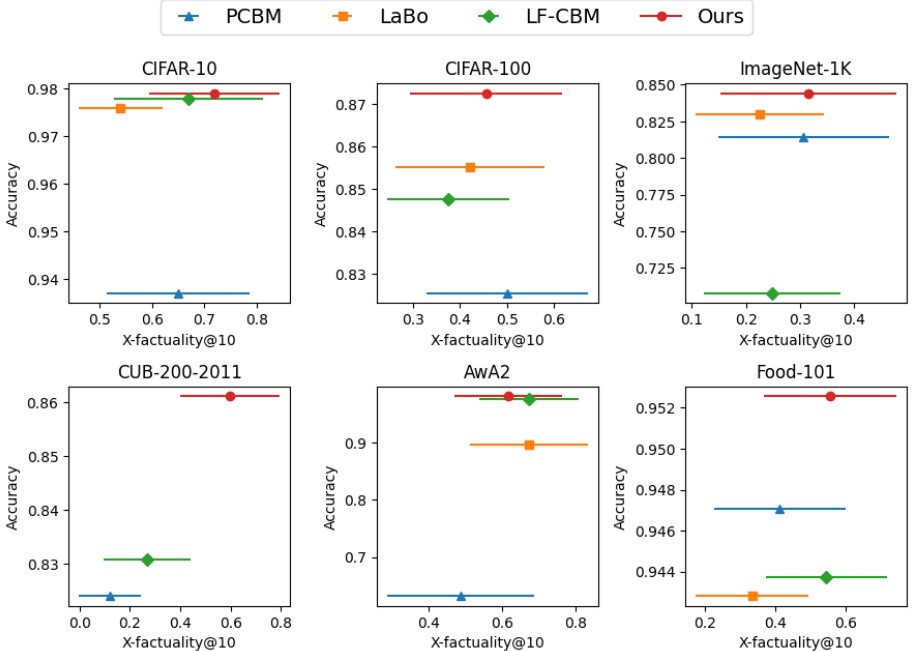

Figure 3: Accuracy v.s. X-Factuality@10 in different datasets. X-Factuality is calculated per class and then aggregated as mean and standard deviation across classes.

count how many are valid. The definition of *validity* of concepts differs per dataset, depending on whether human annotations are present or not:

1. For concepts curated from ConceptNet (CIFAR-10, CIFAR-100, and ImageNet-1K), a concept is *valid* if there exists a valid edge connecting it and the corresponding class name[2].

2. For concepts collected from GPT (Food-101), we decide valid concepts by prompting GPT-4 with "*Please assign a score between 0 and 1 based on the importance of {concept} in visually recognizing {class name}*". A threshold of 0.5 is applied to define valid concepts.

3. For AWA2, human annotations provide class-wise binary concept labels, indicating the presence of concepts in certain classes of animals. We consider concepts with positive labels *valid*.

4. For CUB-200-2011, human annotations assign a binary presence label and an annotator-confidence label ("definitely", "probably", "guessing", or "invisible") for each image-concept pair. Image-wise presence labels are aggregated to class-wise continuous labels, with values between 0 and 100, indicating the percentage of times a concept is marked as "present" for a given class. We use 50% as a threshold and consider concepts with values above that as *valid*.

We denote the set of top $k$ concepts for class $i$ as $S_i^k$ and the "valid" concepts as $S_i^v$. Given $S_i^k$ and $S_i^v$, we define X-factuality as:

$$\text{X-factuality}_i@k = \frac{|S_i^k \cap S_i^v|}{k}, \qquad (8)$$

Note that while the definition of X-factuality seems related to the commonly used metric, precision, they are not equivalent since "valid" concepts are not necessarily important in a prediction task. For instance, "fur" might be "valid" in many animals, but might not be important in classifying different types of animals. A high X-factuality only indicates the selected concepts do not contradict human consensus, but do not guarantee the correctness of an explanation, since the actual importance of these concepts for making predictions is not always clear or well-defined.

---

[2]The ConceptNet assigns a semantic meaning (e.g., "Has A", "Part of") to each edge describing the relationship, and we consider an edge valid if its semantic meaning is not "Obstructed By", "Antonym", "Distinct From", or "External URL".

Table 3: Local Explanation Quality of CUB-200-2011

| Annotations | | Composition of Top-10 Important Concepts (%) | | |
|---|---|---|---|---|
| Presence | Certainty | PCBM | LF-CBM | CBM-zeros (Ours) |
| Yes (↑) | Definitely | 5.95 ± 8.39 | 15.7 ± 15.7 | **34.3 ± 24.5** |
| | Probably | 2.71 ± 6.01 | 6.99 ± 11.2 | **13.3 ± 18.4** |
| | Guessing | 0.59 ± 2.92 | 1.30 ± 4.54 | 2.40 ± 10.6 |
| No (↓) | Definitely | **52.6 ± 33.2** | 40.4 ± 28.4 | 27.2 ± 22.4 |
| | Probably | **23.7 ± 28.2** | 20.9 ± 24.2 | 14.1 ± 18.2 |
| | Guessing | 5.02 ± 13.9 | 5.05 ± 13.1 | 3.52 ± 10.6 |
| Not visible | | 9.45 ± 16.8 | 9.68 ± 13.1 | 5.22 ± 12.2 |

We select the top 10 important concepts per image, calculate their proportions of 7 possible presence-certainty combinations and report the mean and standard deviation across images.

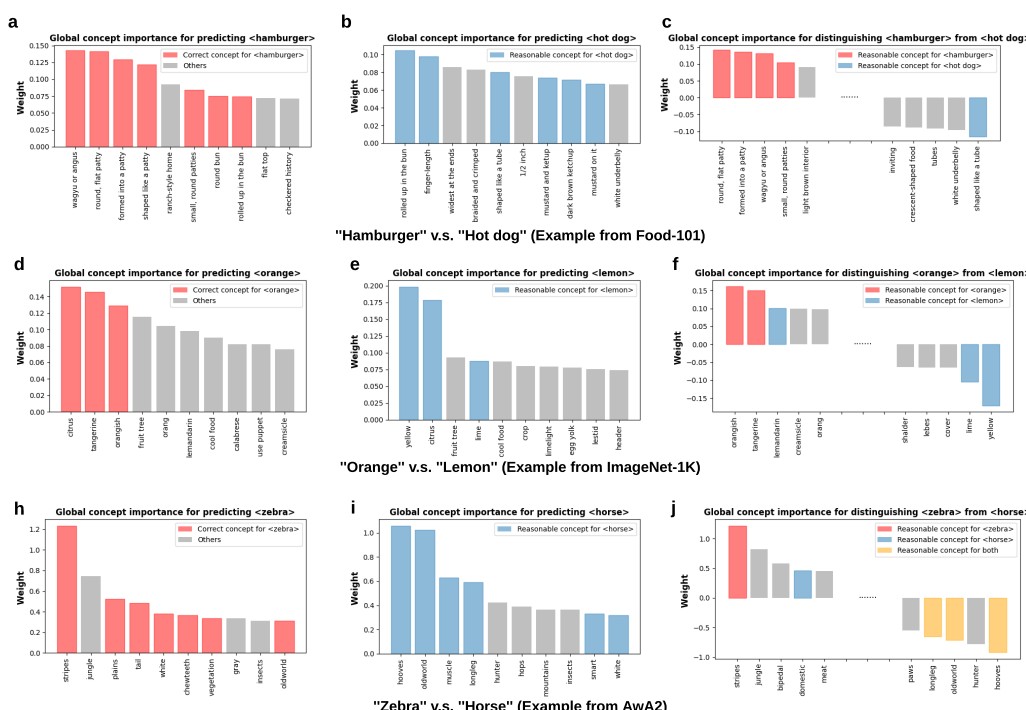

Figure 4: **Examples of global explanation importance. (a-c)** Important concepts for predicting hamburger (a), hot dog (b), and differentiating these two (c). **(d-f)** Important concepts for predicting orange (c), lemon (d), and differentiating these two (f). **(h-j)** Important concepts for predicting zebra (h), horse (i), and differentiating these two (j).

LOCAL CONCEPT CONTRIBUTIONS    The local contributions of concept $s_m$ for predicting the sample $x$ as class $i$ is defined as $\gamma_{i,m}$. Quantitative evaluation of these local explanations needs costly sample-wise concept annotations, which are only available for CUB-200-2011. Here, we select the top 10 concepts with the highest contributions per image and compare to their annotations. For other datasets, we provide qualitative local evaluations.

**Results**    We used CLIP models to generate concept set features $c(x)$ for all the datasets except CUB-200-2011, since CLIP models struggle to capture the fine-grained annotated concepts and severely hurt the faithfulness of explanation for all methods (see more details in Appendix A.6 and Table A.1). Thus, we use human-annotated presence labels as $c(x)$ for this latter dataset. For fair comparisons, we also use ground truth annotations in PCBM (the CAV version) and LF-CBM. Yet, this does not apply to LaBo, as the CLIP score is essential to its input. Table 1 and Table 2 summarize the classification accuracy (ACC) and X-Factuality@10 (X-Fact@10) of global explanation for general and fine-grained tasks, respectively. X-factuality is calculated per class and aggregated as

mean and standard deviation across classes. We also visualize results in 2D panels, with X-factuality on the x-axis and accuracy on the y-axis (see Figure 3). Therefore, points closer to the top-right corner indicate better performance in both predictive power and interpretability. In all but one case, our method is the closest one to the top-right corner. We obtain the highest accuracy across all the datasets (since out method inherently preserves the black-box models' prediction), and the best X-factuality in almost all cases. Figure A.1 further shows X-factuality@$k$ as a function of $k$ across datasets. Moreover, we include some qualitative analysis with representative examples. In Figure 4 we plot the top 10 global concept importance ($\Gamma_{i,m}$) for several pairs of related classes, such as "hamburger" (Figure 4.a) v.s. "hot dog" (Figure 4.b); "orange" (Figure 4.d) v.s. "lemon" (Figure 4.e); and "zebra" (Figure 4.h); "horse" (Figure 4.i). We also analyze how model differentiates class $i$ from class $j$ by plotting $\tilde{A}_{i,m} - \tilde{A}_{j,m}$ (see Figure 4.c, f, j). There are some interesting observations. For instance, shape is a key factor "hamburger" from "hot dog" (round v.s. tube); The key difference between "orange" and "lemon" lies in their colors (orangish or yellow); and "strips" are the key factor separating "zebra" from "horse". These observations align well with our intuitions.

Regarding *local* explanations for CUB-200-2011, we select the top 10 important concepts (ranked by $\gamma_{i,m}$) per image and check their annotations. There are 7 possible presence and certainty combinations in annotations. We calculate their proportions for each image and report the mean and standard deviation across images in Table3. Our method obtains the best alignment with ground truth concept annotations in the test set: among the top 10 important concepts, 34.3% ± 24.5% are annotated as "definitely present", and another 13.3% ± 18.4% are annotated as "probably present", significantly better higher than those in comparative methods. We show examples of local concept contributions for CIFAR-10 (Figure A.5), CIFAR-100 (Figure A.6), Imagenet-1K (Figure A.10, A.11, A.12, A.13), CUB-200-2011 (Figure A.7), AwA2 (Figure A.8), and Food-101 (Figure A.9) in Appendix.

## 5 DISCUSSION, LIMITATIONS, AND CONCLUSION

In this work, we introduce CBM-zero, which explains black-box models by constructing a CBM via an invertible mapping between its latent space and an interpretable concept space. This concept space can be derived from CLIP models, eliminating the need for dense concept annotations. Unlike other CBMs, CBM-zero does not alter the original black-box model, preserving its performance exactly. Experiments across various benchmarks demonstrate its superiority in maintaining the model's accuracy and providing high-quality interpretations compared with states-of-arts.

This work also has several limitations. First, the affine mapping we use might not always be powerful enough to map image features from hidden space to concept space accurately, which could affect the faithfulness of the explanation (Margeloiu et al. (2021); Havasi et al. (2022); Huang et al. (2024)). Leveraging more expressive yet still invertible models (e.g., normalizing flows Kobyzev et al. (2020)) in the future might address this. Moreover, the reliance on CLIP models introduces limitations when concepts are fine-grained features (e.g., CUB-200-2011 annotations) without fine-tuning. CLIP scores measure correlation rather than directly indicating the presence of concepts. Therefore, an image of a baby might get a high score for "stroller" even if there is no stroller shown in the image. Exploring alternatives, such as querying large language and vision models (Chattopadhyay et al. (2024)) or human collaborators (Chauhan et al. (2023)) about the presence of certain concepts, might address this issue. Finally, the concept banks used are not perfect. Concept-Net, although large and evolving, is not sufficient to represent the numerous concepts and complex relationships in the real world. GPT-generated concepts are more flexible and diverse but might be overly complex and sometimes these concepts are not visual descriptions. Both ConceptNet and GPT-generated concepts can be inaccurate in describing the relationship between concepts, with unclear levels of noise. Human annotations, while more accurate, are costly and labor-intensive, hindering its board applicability. Incorporating recent advances in concept discovery (Shang et al. (2024); Huang et al. (2024); Schrodi et al. (2024); Hu et al. (2024)), as well as incorporating notions of uncertainty quantification to our results (Angelopoulos et al. (2020); Teneggi et al. (2023)), may help generate more reasonable explanations.

## 6 REPRODUCIBILITY STATEMENT

Codes for implementing this method and reproducing the results can be found in this anonymous repository: https://anonymous.4open.science/r/CBM-zero-EBED

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

# A APPENDIX

## A.1 APPENDIX OVERVIEW

The Appendix provides additional clarifications and details that are not included in the main text due to the length limit. Section A.2 shows X-factuality@$k$ as a function of $k$. Section A.3 provides quantitative results supporting the claim that "CLIP models struggle to culture the fine-grained annotated concepts from CUB-200-2011", as mentioned in Section 4. In Section A.4, we provide implementation details concerning concept curation and selection procedure and computational efficiency of this work. Section A.5 discusses the sensitivity of results to hyperparameters, exponential power $t$, and regularization strength $\lambda$, as well as the impact of different versions of CLIP models. Section A.6 includes examples of local explanations.

## A.2 X-FACTUALITY@$k$ AS A FUNCTION OF $k$

In the main text, we only report X-factuality @ 10 for global explanations in Table 1 and 2. Here, we extend this by plotting X-factuality@$k$ as a function of $k$, as shown in Figure A.1.

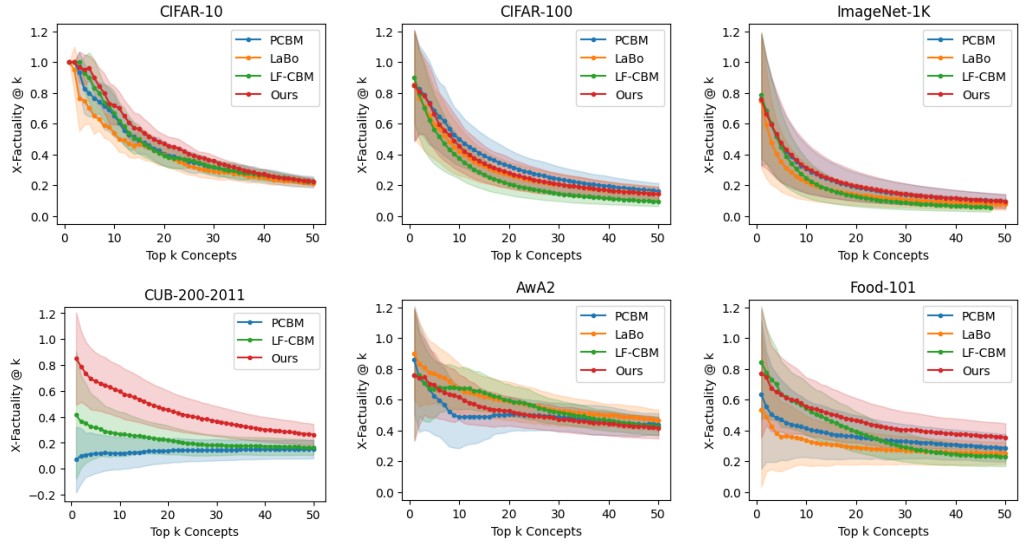

Figure A.1: X-Factuality @ $k$ v.s. $k$ in CIFAR-10 (a), CIFAR-100 (b), ImageNet-1K (c), CUB-200-2011 (d), AwA2 (e), and Food-101 (f). Solid lines show the mean X-Factuality across classes, and the shaded area shows the standard deviation among classes.

## A.3 CLIP MODELS ON CUB-200-2011

As described in Section 4, the CUB-200-2011 dataset contains 312 human-annotated concepts. These concepts are fine-grained descriptions of the color, shape, and size of specific bird parts. We found that even state-of-the-art CLIP models struggle to align these concepts with the correct images, reducing the faithfulness of any explanation methods relying on them. To demonstrate this, we select the top 10 concepts with the highest CLIP scores per image and compare them with annotations. Table A.1 summarizes the results of different versions of CLIP models, and none of them align with the ground truth well.

## A.4 IMPLEMENTATION DETAILS

**Concept curation and selection** For CIFAR-10, CIFAR-100, and ImageNet-1K, we collect concepts by querying ConceptNet (Speer et al. (2017)) with the {*class name*} and obtain {*concepts*}). Only concepts with valid connections (excluding connections with semantic meaning of "Obstructed

Table A.1: Quality of different versions of CLIP models on capturing concepts of CUB-200-2011

| Annotations | | Composition of Top-10 Correlated Concepts (%) | | | | |
|---|---|---|---|---|---|---|
| Presence | Certainty | RN50 | RN101 | ViT-B/16 | ViT-B/32 | ViT-L/14 |
| | Definitely | 7.71 ± 11.5 | 6.69 ± 10.5 | 8.97 ± 13.0 | 8.45 ± 11.8 | **10.4 ± 12.3** |
| Yes(↑) | Probably | 4.93 ± 9.04 | 3.73 ± 7.92 | 5.04 ± 9.74 | 5.43 ± 9.76 | **5.86 ± 9.72** |
| | Guessing | 1.43 ± 4.43 | 0.97 ± 3.73 | 1.17 ± 4.41 | 1.54 ± 4.93 | 1.44 ± 4.60 |
| | Definitely | 39.3 ± 31.6 | **45.6 ± 33.7** | 41.0 ± 32.1 | 35.7 ± 32.8 | 38.1 ± 31.6 |
| No(↓) | Probably | 23.0 ± 27.1 | **25.4 ± 29.0** | 23.1 ± 27.3 | 21.7 ± 27.8 | 21.8 ± 27.5 |
| | Guessing | 6.37 ± 15.3 | 6.10 ± 15.4 | 5.66 ± 14.6 | 6.25 ± 16.1 | 6.05 ± 15.9 |
| Not visible | | 17.3 ± 24.0 | 11.6 ± 18.7 | 15.1 ± 22.5 | 20.9 ± 29.6 | 16.4 ± 25.8 |

We select the top 10 concepts with the highest CLIP scores per image, calculate their proportions of 7 possible presence-certainty combinations, and report the mean and standard deviation across images.

By", "Antony", "Distinct From", or "External URL") are retained. For a specific dataset, the concepts connecting all the class names are gathered, and the following processing is applied: (1) long concepts with more than 10 characters are excluded to include simple concepts instead of complex statements; (3) only the top 10 concepts with the highest connection strength to each class name are preserved, which excludes those less common, and weakly related concepts; (3) concepts that are close to each other and close to class names are excluded, where the text-text similarity is measured by cosine similarities of the embedding encoded by *all-mpnet-base-v2* (Reimers (2019)) sentence encoder and a threshold of 0.85 is applied. For Food-101, we use the concepts curated by LaBo (Yang et al. (2023)), where the GPT-3 is prompted with *describe what {class name} looks like* and relevant concepts are extracted from the answers. We exclude overly long and complex concepts with more than 15 characters.

**Computational efficiency** All the models are trained on a single Nvidia GPU. The training of a black-box model takes from a few minutes to two days depending on the dataset size. Once the feature embedding of the black-box model and CLIP image encoders are saved, the training of the affine mapping for interpretation purposes is typically within 10 minutes.

A.5    HYPERPARAMETER SENSITIVITY AND ABLATION

As described in Section 3.2, we apply an exponential transformation with power $t$ on the CLIP scores to emphasize concepts with higher correlation to images. In the optimization function (equation 3), $\lambda$ controls the regularization strength. In this section, we discuss the sensitivity of global explanation quality to these two hyperparameters. Moreover, we explore the impact of using other versions of CLIP models. Note that the classification accuracy remains unchanged given the same black-box model, regardless of the choice of hyperparameters and CLIP models.

**Exponential power $t$** Figure A.2 shows X-factuality@$k$ as a function of $k$ with exponential transformation power $t$ of 1 (i.e., no exponential transformation), 3, and 5 across different datasets. In all cases, $t = 5$, gains the best results, particularly for Food-101.

**Regularization strength $\lambda$** Figure A.3 shows X-factuality@$k$ v.s. $k$ of global explanation for different regularization strengths $\lambda$. The default choice, $\lambda = \frac{2}{d}$ in general performs best across different sets, and the results are not sensitive to the choice of $\lambda$. Data-specific hyperparameter tuning is expected to boost the results further.

**Other versions of CLIP models** In the main text, we use CLIP-ViT-L/14 by OpenAI as the CLIP model to construct the concept bottleneck models. Here, we explore the results of using other versions of CLIP: RN50, RN101, ViT-B/32, and ViT-B/16. Figure A.4 shows the X-factuality for different models, with CLIP-ViT-L/14 consistently obtaining the best results.

A.6    EXAMPLES OF LOCAL EXPLANATIONS

In this section, we provide some examples of contributions of concepts in the prediction of specific images across different datasets. To be more specific, we focus on the deviation of logit of the

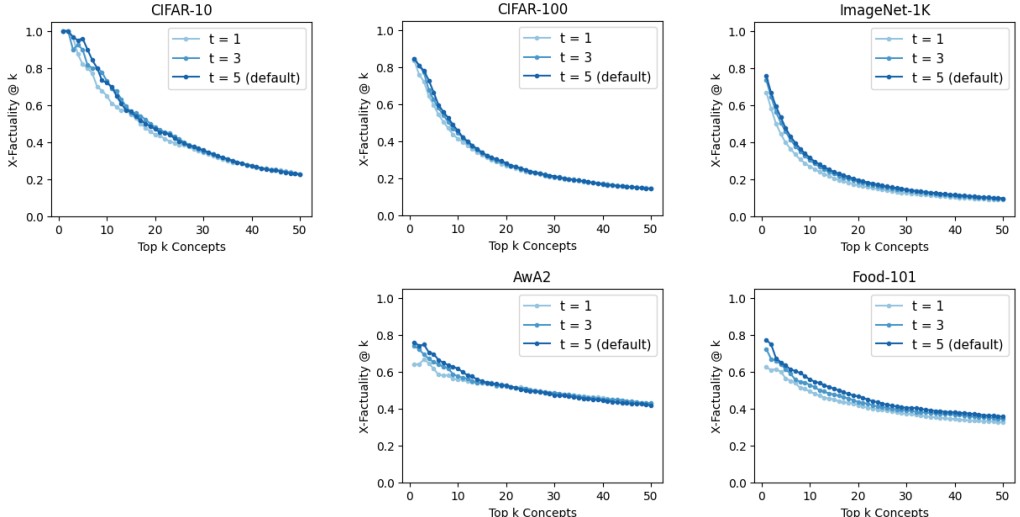

Figure A.2: Global explanation quality of using different exponential power $t$. X-factuality@$k$ v.s. $k$ are shown. Results of CUB-200-2011 are not included, as binary annotated labels are used as $c(x)$, and exponential transformation does not change it.

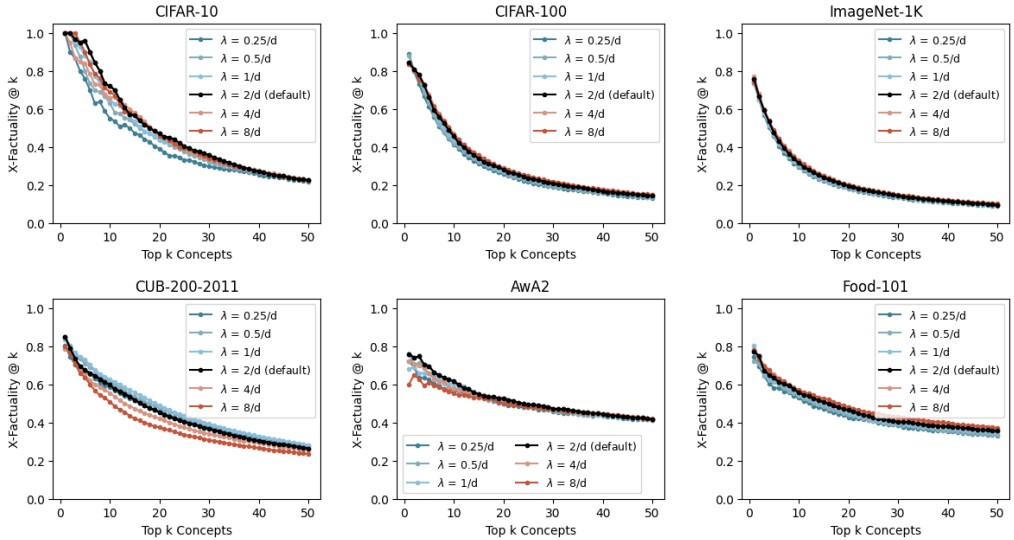

Figure A.3: Global explanation quality of using different regularization strength $\lambda$. X-factuality@$k$ v.s. $k$ are shown.

predicted class from the mean logit across classes, $z_{\hat{y}} - \frac{1}{K}\sum_{i=1}^{K} z_i$, and calculate the contribution of concept $s_m$, denoted as $((AW^+)_{\hat{y}m} - \frac{1}{K}\sum_i (AW^+)_{im})(f(x)+h)_m$ in percentage. The top 10 contributed concepts for each image are shown. Figure A.5 and Figure A.6 show examples from CIFAR-10 and CIFAR-100. Figure A.7 shows examples from CUB-200-2011, Figure A.8 shows examples from AwA2, and Figure A.8. For ImageNet-1K with a lot of images from diverse classes, we show examples by category, furniture in Fig A.10, animals in Fig A.11, clothes in Fig A.12, and locations in Fig A.13.

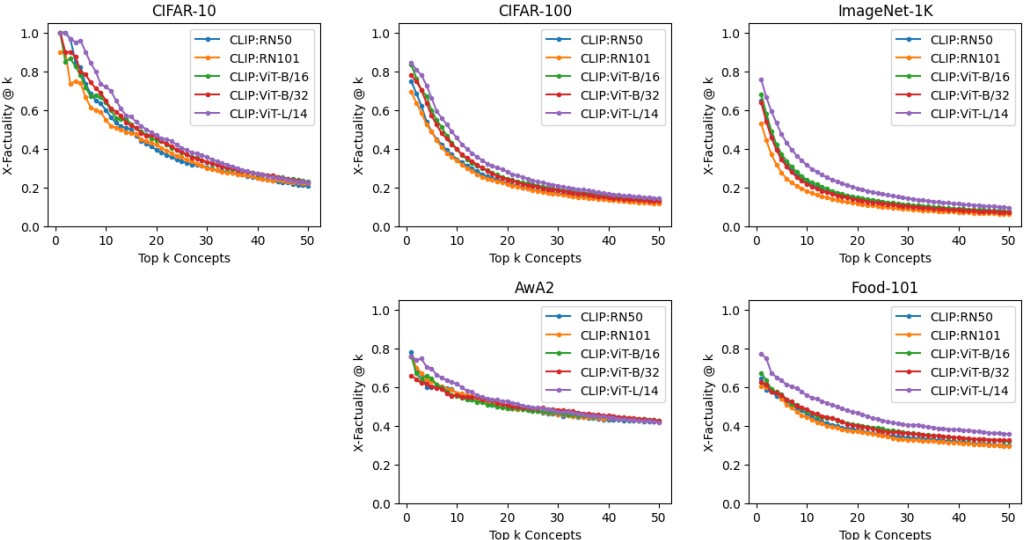

Figure A.4: Global explanation quality of using different versions of CLIP models. X-factuality@$k$ v.s. $k$ are shown. Results of CUB-200-2011 are not included, as binary annotated labels are used as $c(x)$.

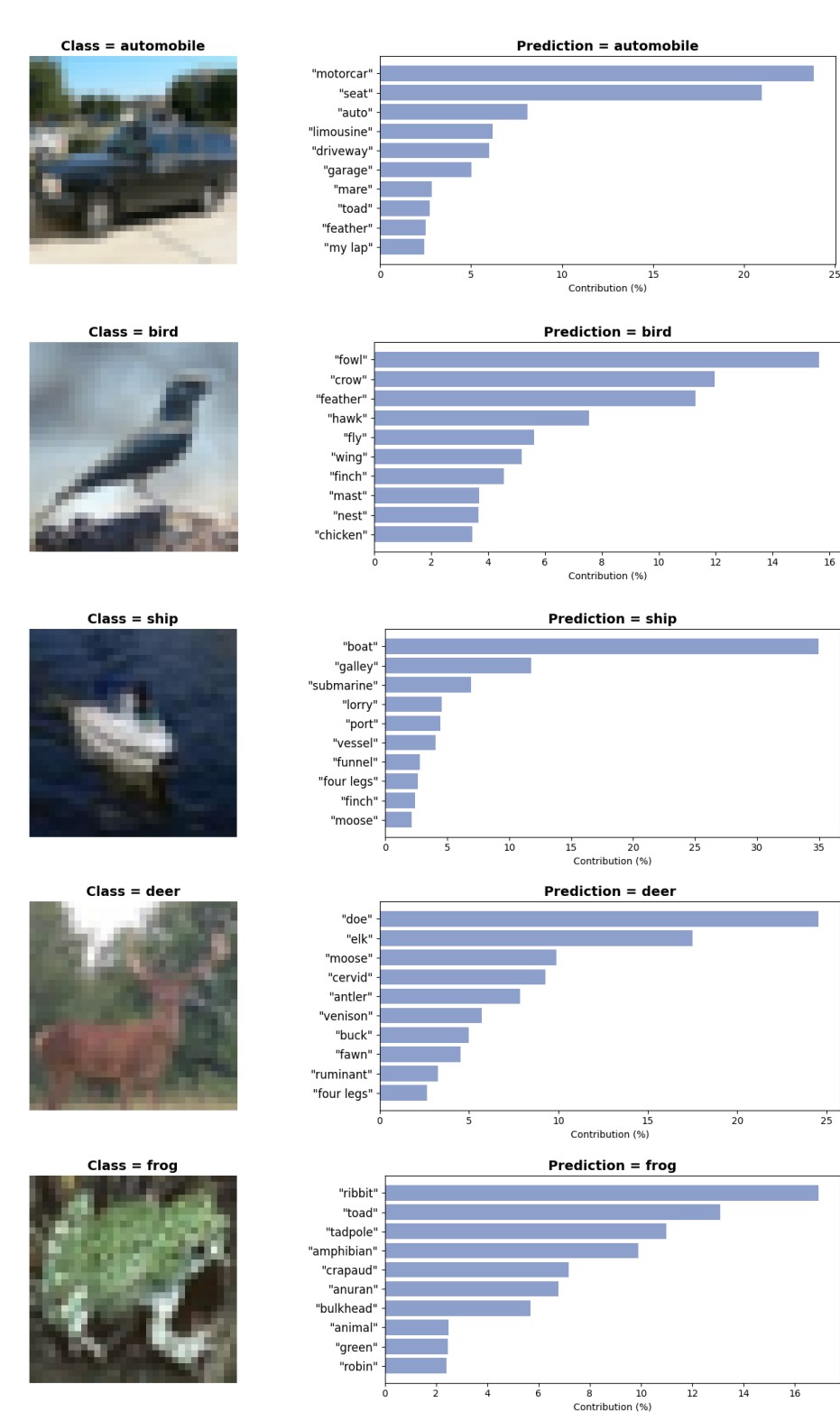

Figure A.5: Examples of local explanations from CIFAR-10. The top 10 contributed concepts are shown.

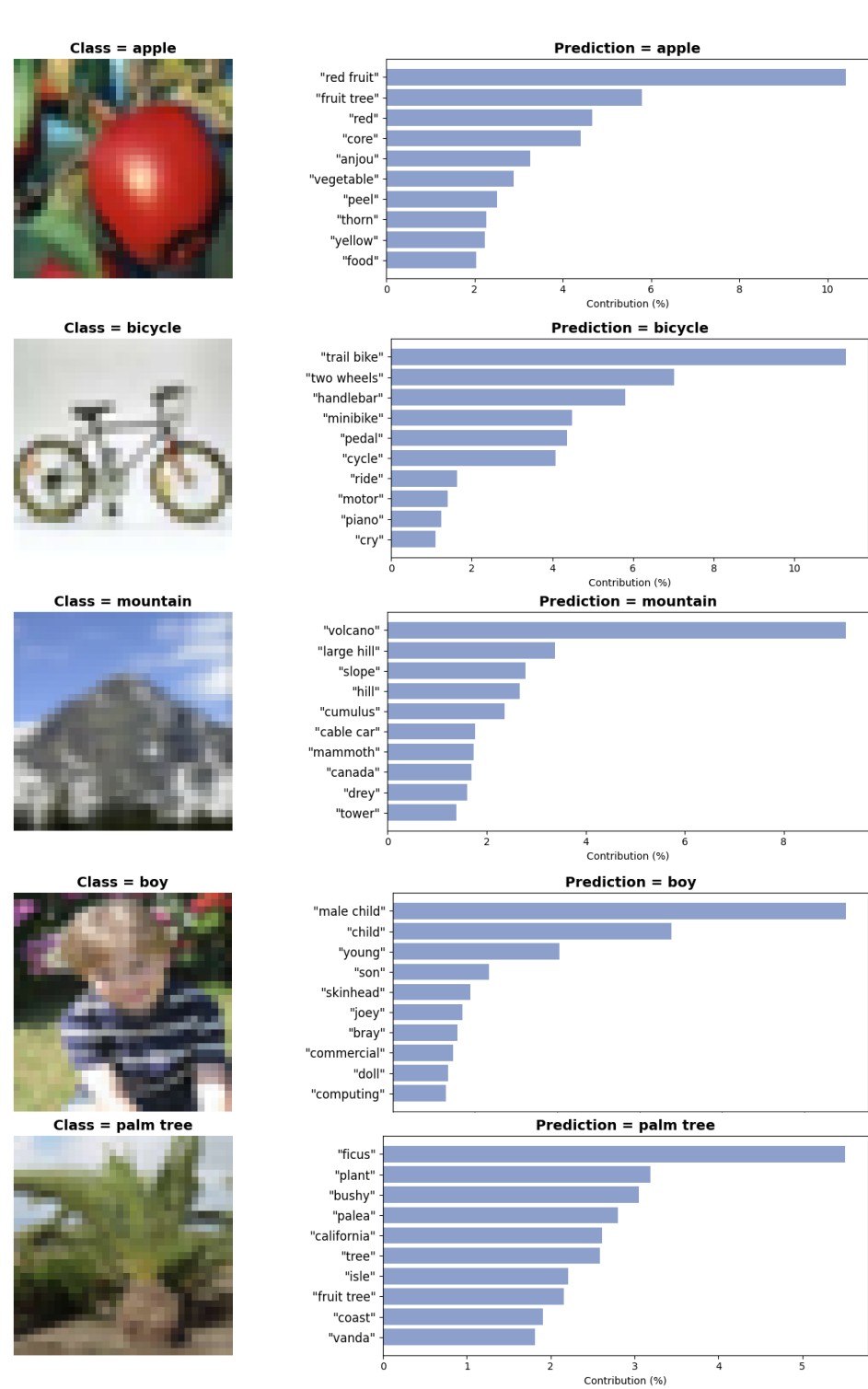

Figure A.6: Examples of local explanations from CIFAR-100. The top 10 contributed concepts are shown.

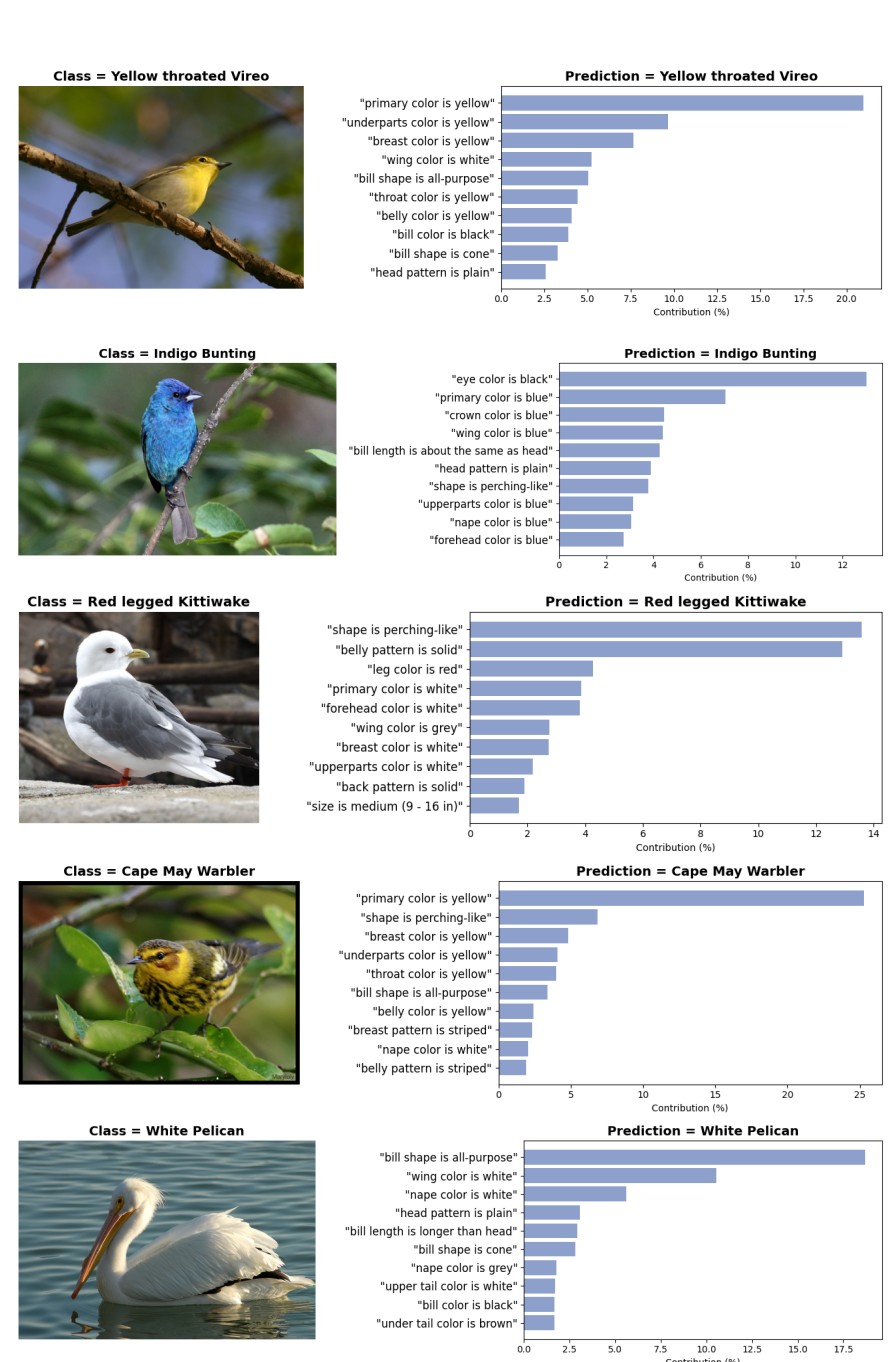

Figure A.7: Examples of local explanations from CUB-200-2011. The top 10 contributed concepts are shown.

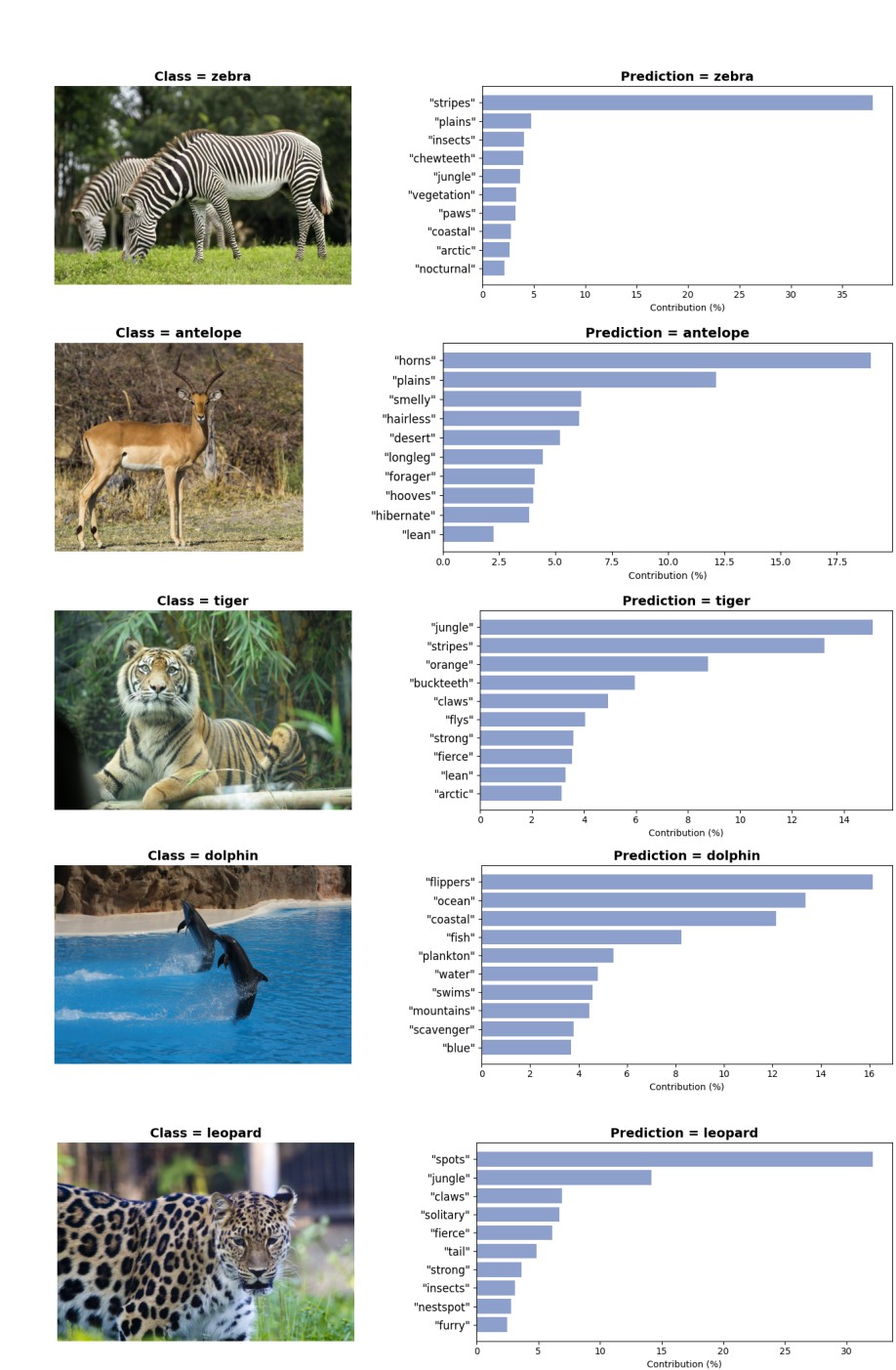

Figure A.8: Examples of local explanations from AwA2. The top 10 contributed concepts are shown.

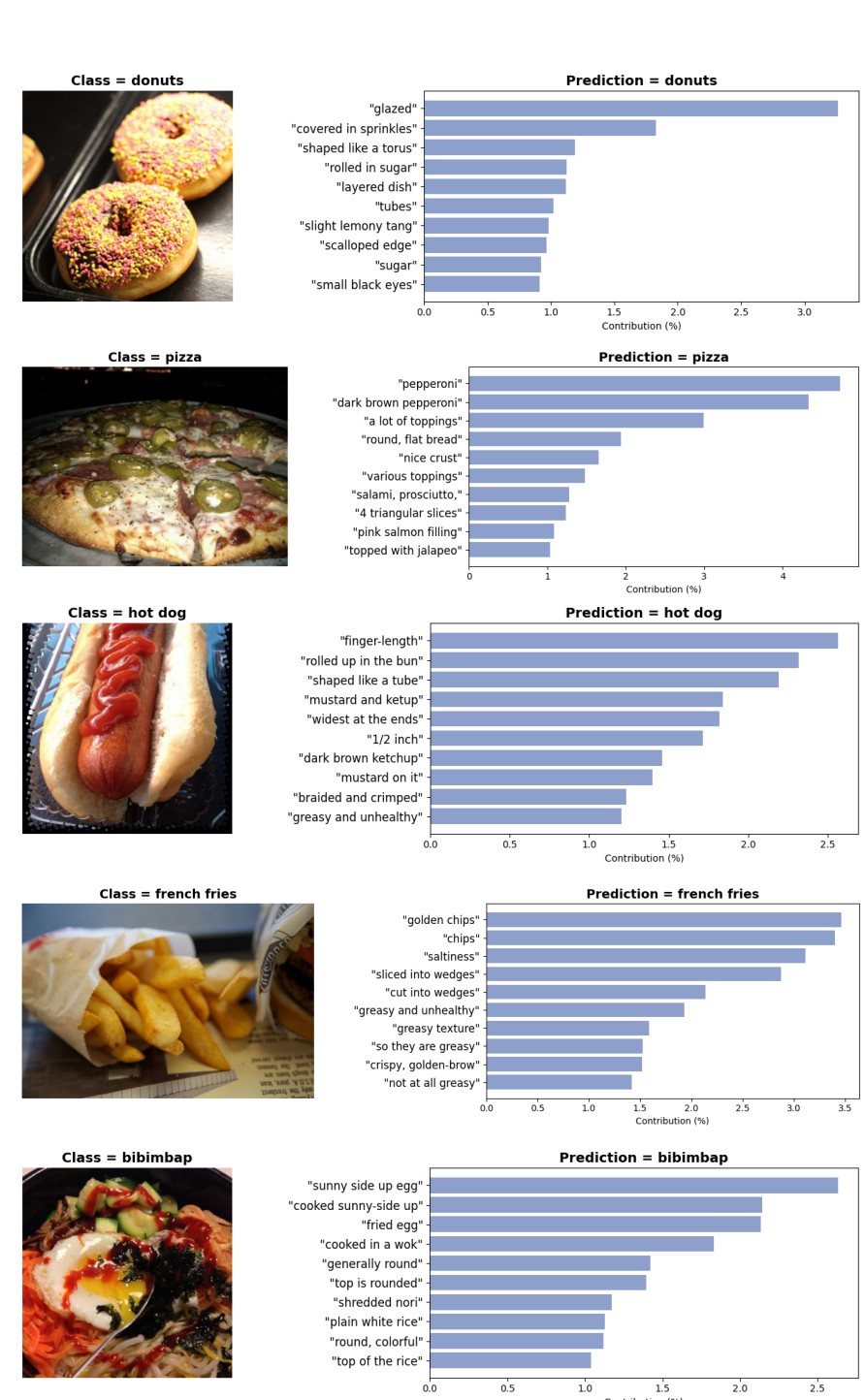

Figure A.9: Examples of local explanations from Food101. The top 10 contributed concepts are shown.

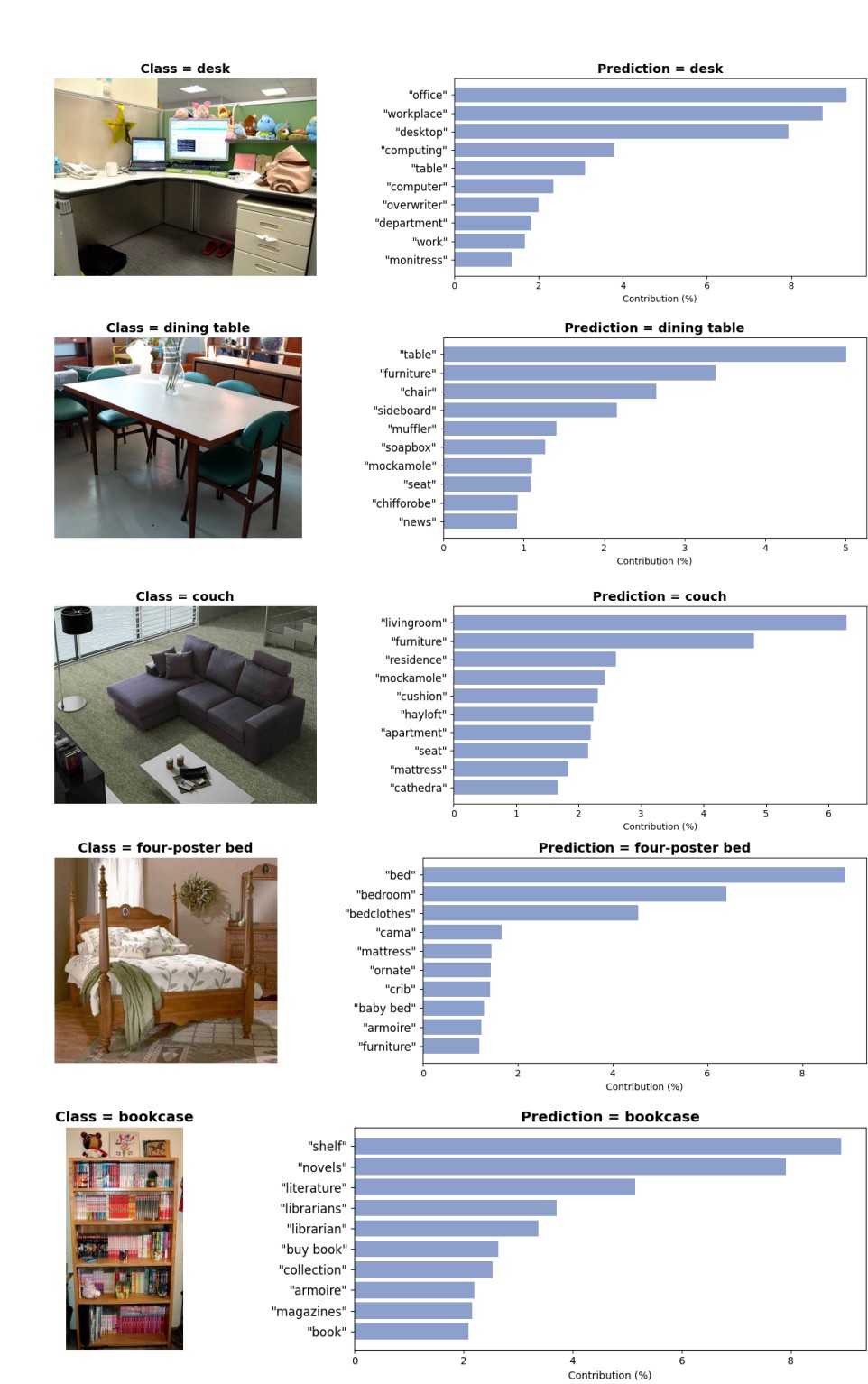

Figure A.10: Examples of local explanations from ImageNet-1K (Furniture). The top 10 contributed concepts are shown.

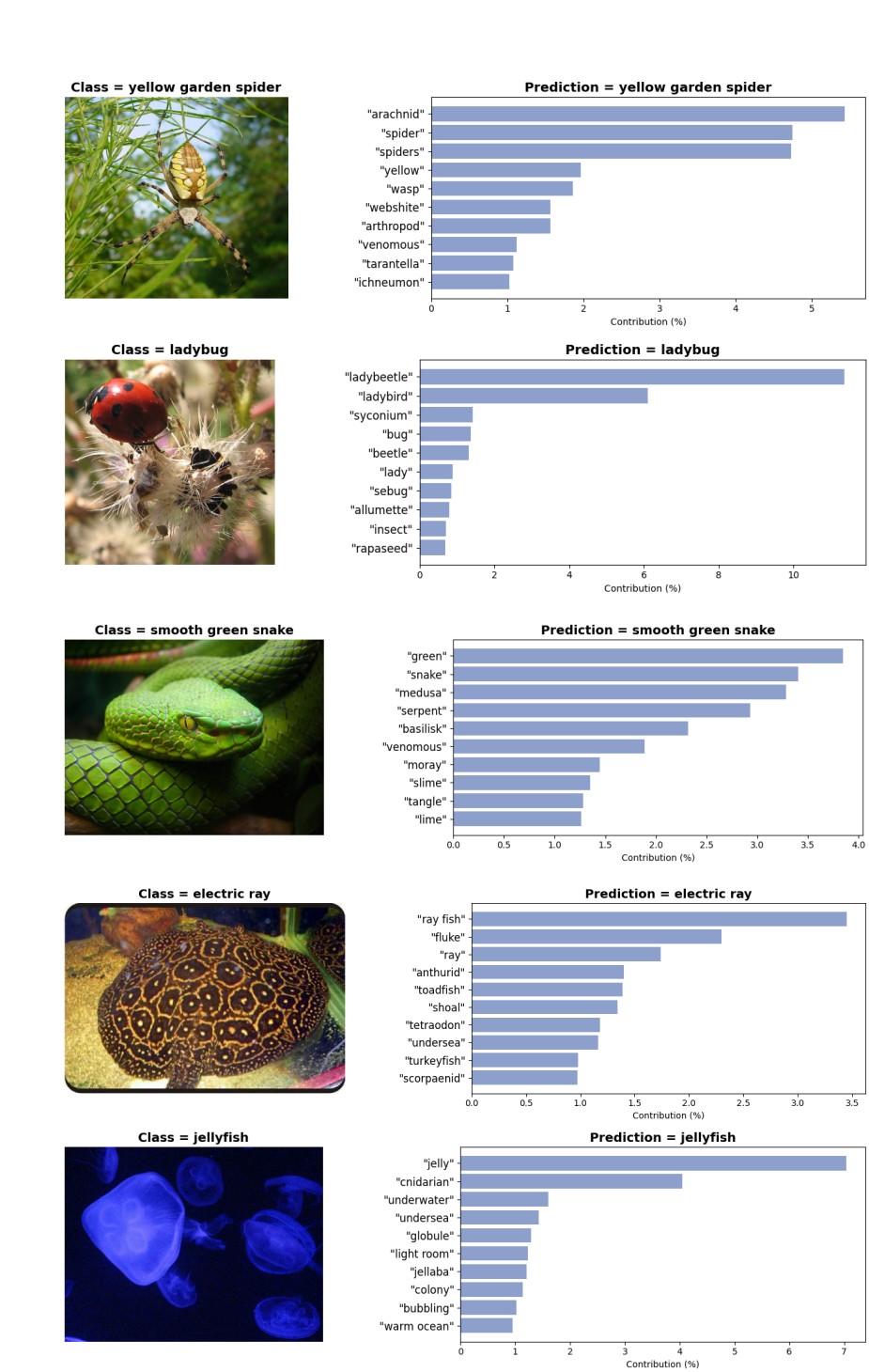

Figure A.11: Examples of local explanations from ImageNet-1K (Animals). The top 10 contributed concepts are shown.

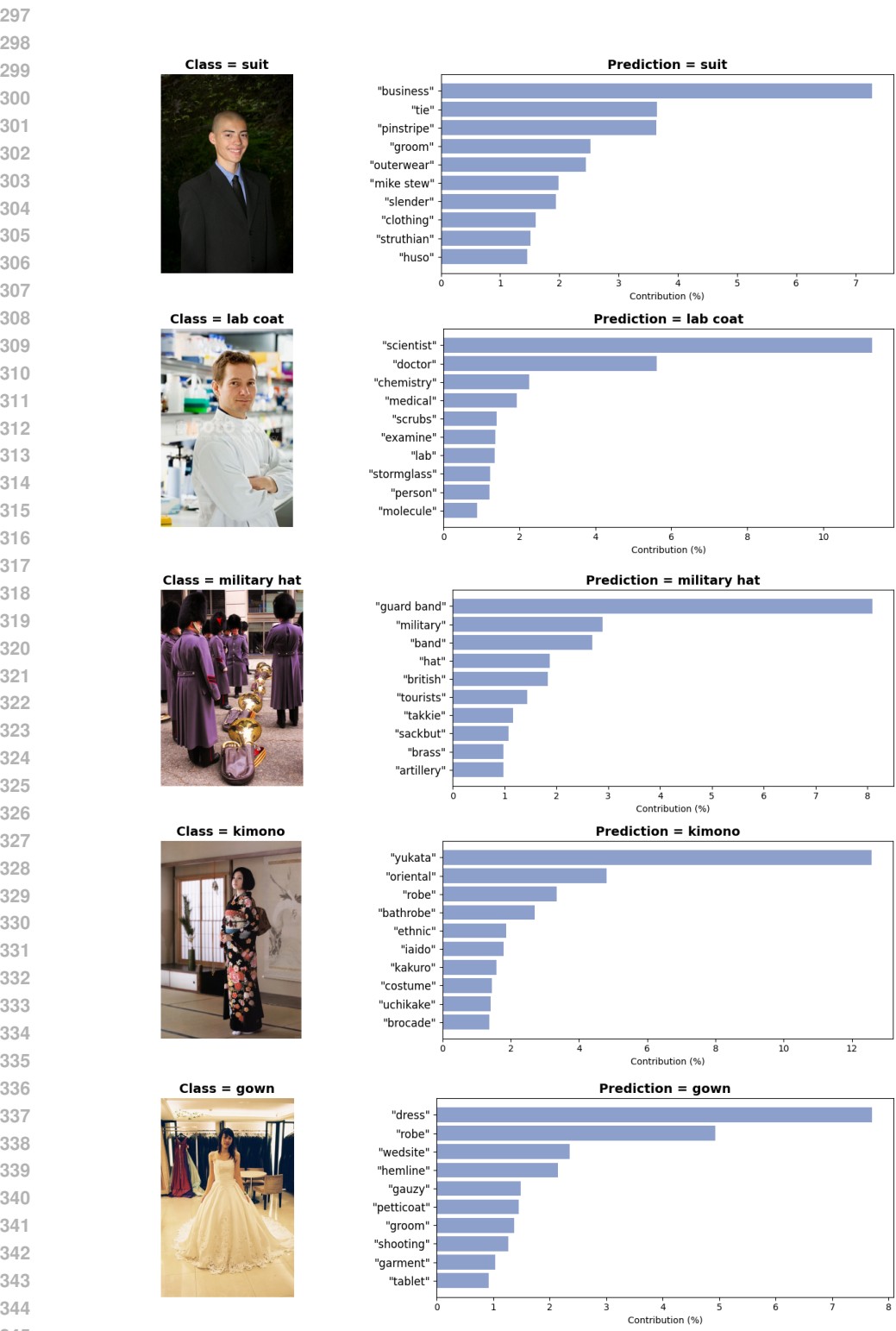

Figure A.12: Examples of local explanations from ImageNet-1K (Clothes). The top 10 contributed concepts are shown.

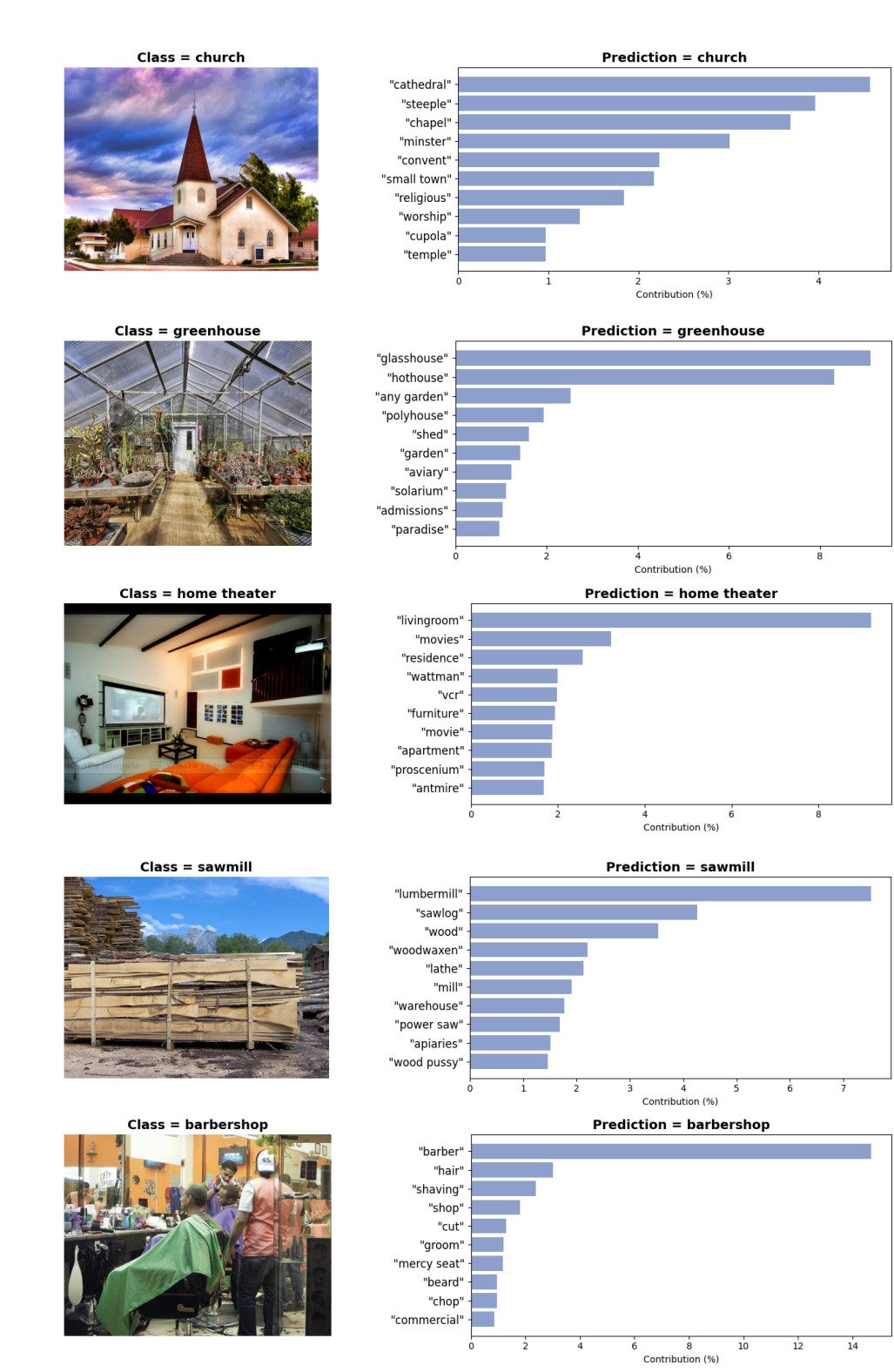

Figure A.13: Examples of local explanations from ImageNet-1K (Locations). The top 10 contributed concepts are shown.

