# OpenReview forum: "CBM-zero: Concept Bottleneck Model With Zero Performance Loss"
_ICLR.cc/2025/Conference — ICLR 2025 Conference Withdrawn Submission_

### Official Review · Reviewer_BN2d · 2024-10-29

**Soundness:** 2
**Presentation:** 3
**Contribution:** 1
**Rating:** 3
**Confidence:** 4

**Summary:**

This paper presents a method to convert black-box models to concept bottleneck models without performance loss. Evaluated on multiple datasets, it shows good accuracy and interpretability compared to other methods.

**Strengths:**

The article proposes a method that enables model interpretability without compromising model performance.

**Weaknesses:**

1. I think this method is somewhat redundant; the explainable part does not directly serve as a basis for classification decisions. In other words, there is no connection between the classification decision and the concept feature, making the explanation meaningless. When you observe an incorrect concept, you cannot rely on human experience to modify the concept to change the prediction result.

2. The innovation is relatively lacking; it merely maps the concept feature back to the backbone feature based on LBF-CBM.

3. The selection of hyperparameters in the experimental section is not explained, such as 𝑡，λ, etc.

4. A notation table for symbols could be added.

**Questions:**

See weaknesses.

---

> ### Author Response · Authors · 2024-11-27
> **Authors' response to Reviewer BN2d**
>
> - Reply to [Weakness 1]: We respectfully disagree with this comment.  In CBM-zero, the decision rule directly (i.e. functionally) depends on the concept features, unlike what the reviewer mentions. Therefore, a manual change in these scores (e.g., as per human interventions) will directly induce a change in the predicted response. We will incorporate new experiments demonstrating these human interventions.
> - Reply to [Weakness 2]: Both LBF-CBM and CBM-zero map backbone features to concept space, but no method has been shown to do this without modifying the original predictor. Our method enforces the invertibility of the mapping, enabling interpretations without performance trade-offs.
> - Reply to [Weakness 3]: We have discussed the sensitivities of these hyperparameters in Appendix A5. We also clearly referenced in the main text that `` We discuss other choices for these parameters and their sensitivity in Appendix A.5, specifically in Figure A.2 and A.3 (lines 366-367)''.
> - Reply to [Weakness 4]: Thank you for your suggestion. We will consider adding a notation table.

---

### Official Review · Reviewer_D9U7 · 2024-11-03

**Soundness:** 3
**Presentation:** 3
**Contribution:** 1
**Rating:** 3
**Confidence:** 5

**Summary:**

This paper presents CBM-Zero, a concept bottleneck model (CBM) that transforms arbitrary black-box models into interpretable ones while preserving their accuracy. To do this, CBM-Zero solves a regression task to learn the mapping between the feature of the black-box model and concept vectors generated from the CLIP encoders. To preserve the accuracy of the original model, CBM-Zero constrain full-rankness of the regression weight matrix so that the transformation by the weight parameters is invertible at the inference time. Experiments show that CBM-Zero achieves high concept prediction performance while preserving the accuracy of the black-box model by comparing it to the black-box model and the CBM baseline model using CLIP.

**Strengths:**

- This paper shows that the proposed method can construct an interpretable model by training projections from to concept space to satisfy full-rankness in CLIP-based CBMs without the performance degradation from the original black-box model.
- The optimization algorithm for satisfying the full-rankness of the regression weight parameters proposed by the paper is solid and has a theoretical background to guarantee accuracy.

**Weaknesses:**

- The matrix weight $W$, which represents the transformation from black-box features to concept space satisfying full-rankness, provides more degrees of freedom for solving the concept regression task, and the concepts selected by $W$ may be dense. Dense concepts are difficult for humans to interpret [a] and may select essentially irrelevant concepts as a result of the optimization to recover the feature vector. The paper introduces a regularization term to induce sparsity but does not evaluate the extent to which this term induces sparsity. Furthermore, in Table 3, the proposed method performs better than the other baselines in predicting concepts that should be present in the image (the rows of "Presence Yes") but poorly in predicting concepts that should not be present (the rows of "Presence No"). This suggests that the proposed method may be assigning higher scores to irrelevant concepts.
- The impact of motivation of this paper is no longer small; the problem of the degrading accuracy of CBMs has already been discussed in an existing study [b]. In contrast to this paper, which aims to preserve the accuracy of the black box model, the existing method achieves performance that outperforms the original model. In this sense, the maximum accuracy achieved by the proposed method is the original black-box model and thus has little impact on the research area.
- The paper introduces an MLP that includes nonlinear transformations to deal with a large number of concepts in the ImageNet-1K experiment (L359). Since the MLP is a black box model inherently, using the MLP to predict concepts contradicts to the purpose of obtaining interpretability. Also, the fact that the proposed method cannot be optimized unless the concept set introduces MLP can be an important limitation of the proposed method. To truly assess interpretability, training results in the linear layer should be evaluated.

[a] Ramaswamy, Vikram V., et al. "Overlooked factors in concept-based explanations: Dataset choice, concept learnability, and human capability." Proceedings of the IEEE/CVF Conference on Computer Vision and Pattern Recognition (CVPR). 2023.

[b] Rao, Sukrut, et al. "Discover-then-name: Task-agnostic concept bottlenecks via automated concept discovery." European Conference on Computer Vision (ECCV). 2024.

**Questions:**

Please see the weakness section and clarify the concerns.

---

> ### Author Response · Authors · 2024-11-27
> **Authors' response to Reviewer D9U7**
>
> - Reply to [Weakness 1]: Thank you for your insightful comments. We will expand on the reported evaluations to show how regularization indices sparsity. We want to clarify that CBM-zero also performs better than other baselines in predicting concepts that should not be present. In Table 3, the rows of "Presence No'' show the percentage of top-10 important concepts that are annotated as ``Not present''. Thus, lower values indicate better results.
> - Reply to [Weakness 2]:   Thank you for referencing this very recent paper. As far as we understand, the CBM models created in that reference outperform the zero-shot classifiers but underperform linear probes in most cases. Note that the linear probe is a very simple classifier (i.e., a linear classifier built on an established encoder). Naturally, the performance can be easily boosted even with slightly more complex models and task-specific fine-tuning. We are not sure whether other CBMs can achieve similar or better performances in these cases, but CBM-zero is guaranteed to preserve the performance.
> - Reply to [Weakness 3]: Thank you for your comment. We want to clarify that the affine and inverse mapping process is applied to all the experiments across different datasets. The MLP is used only in constructing black-box models, not as part of the interpretability process. We will revise the text to make this clearer.

---

### Official Review · Reviewer_Ccrw · 2024-11-03

**Soundness:** 1
**Presentation:** 1
**Contribution:** 2
**Rating:** 1
**Confidence:** 5

**Summary:**

The authors propose CBM-zero, a variant of CBM that utilizes a learnable matrix to map features from a bottleneck layer to concept representations. The matrix is ensured to be invertible and is subsequently utilized to map the concept representations back to the latent space and reuse the original classification layer to perform the prediction. In addition, the authors have utilized augmented CLIP scores as concept annotations and have also proposed the factuality metric to judge performance.

**Strengths:**

1. The paper has a good experimental section.
2. The metric of Factuality might be useful.

**Weaknesses:**

1. Wrong understanding of CBMs: In my opinion, the authors have misunderstood the intention of proposing CBMs. CBMs are not merely concept-mappers but should support both concept-predictions and $interventions$ - i.e. changing a wrong concept value should "fix" or "correct" the prediction. Please note that all papers cited by your work [1,2,3] have dedicated sections for concept interventions (usually the last section) while CBM-zero does not. Without an intervention result, any CBM architecture is useless and is just a multi-task network with a prediction head and a concept head.
2. Lack of coherence across concept sets: Authors utilize [1,2,3] as baselines to compare their approach. However, all these approaches use vastly different concept sets and there is no common methodology to generate concepts - making comparisons meaningless. One way to properly compare the results is to use the same concept generation methodology and utilize similar concepts to measure their influence/contribution, etc. In addition, there should be a concept-error (performance) metric [Refer to the CBM paper] for a more fair evaluation. In its current state, the methodology's efficacy is not established. If the authors feel the concept set's performance is truly remarkable - a human study should also be conducted.
3. No loss of performance is a bug, not a feature: As I mentioned before, without a similar concept set comparison to other approaches is not meaningful. As the concepts are themselves not evaluated to be meaningful, there is no surprise the performance of a black box approach and CBM-zero is the same.

(Suggestions - you are welcome to follow these or not)
1. The work is wrongly positioned between post-hoc and interpretable-by-design CBM approaches. In my opinion, the work should be in the latter category and only compared against LF and LaBo approaches.
2. Concept-set standardization: Please only use either GPT-generated concepts or ConceptNet concepts in your evaluation. If necessary two separate evaluations can be done to ascertain which approach works better.





[1] Post-hoc CBMs, ICLR 23

[2] Label-free CBMs, ICLR 23

[3] Language in a Bottle, CVPR 23

**Questions:**

See Weakness.

---

> ### Author Response · Authors · 2024-11-27
> **Authors' response to Reviewer Ccrw**
>
> - Reply to [Weakness 1]: We agree that intervention is an important and appealing feature of CBMs. The proposed CBM-zero also supports intervention, as we can fix the values in $Wf(x)+h$ to intervene on and improve the black-box model. We will include experiments to demonstrate its intervention ability. However, we do not agree with the reviewer that CBMs without intervention are useless. CBMs (even without intervention) provide high-level interpretations for the prediction, which can inform model debugging. Finally, note that [3] does not include any intervention experiments.
> - Reply to [Weakness 2&3]: Indeed, we use the same concept set when comparing different approaches on each individual dataset, as we clearly stated in line 353, ``All methods use the same concept bank per dataset''.  We used different concept-generation strategies for different datasets to account for task-specific differences. Some datasets (awa2, cub) have concept annotations and ground truth with them, and others do not. Some datasets are general classification tasks with common class names (CIFAR10, CIFAR100) that are included in ConceptNet, but others are fine-grained tasks with specific names (Food101). We thank the suggestion to add a concept-error metric for a more comprehensive evaluation.

---

### Official Review · Reviewer_RYAn · 2024-11-03

**Soundness:** 2
**Presentation:** 3
**Contribution:** 3
**Rating:** 5
**Confidence:** 3

**Summary:**

A primary limitation of Concept Bottleneck Models (CBMs) is their lower accuracy than conventional black-box models due to their reliance on a surrogate predictor rather than the original model. To address this issue, the authors introduced CBM-zero, a novel CBM that can be integrated with any standard black-box model through an invertible mapping from its latent space to an interpretable concept space. Experimental results demonstrated that relative to other label-free CBM approaches, the proposed model performs comparably to black-box models, which is a favorable outcome.

**Strengths:**

- S1: The proposed method, along with the definitions of global and local explanations, is clearly articulated and intuitive.

**Weaknesses:**

- W1 (The constraints of W and its connection with concept sparsity): One of my primary concerns is that the weight matrix \( W \) is mandated to have the number of concepts \( M \) exceed the dimensionality \( d \), which raises issues regarding concept sparsity. This weight constraint imposes significant limitations on the method's capacity for concept sparsity, particularly given that \( d \) in the hidden layers of contemporary AI architectures is typically substantial. Although the authors have introduced a regularizer in Equation 4, it is concerning that the regularizer and the weight constraint may counteract each other's learning processes; therefore, an ablation study is needed to evaluate the effectiveness of the regularizer. Furthermore, Algorithm 1 appears to discourage sparsity by introducing minor perturbations, which further undermines the function of the regularizer.

- W2 (Insufficient Details in Experimental Setups): In line 360, the authors indicate that for ImageNet-1K, an additional Multi-Layer Perceptron (MLP) was utilized due to the extensive size of the concepts. This raises the question of whether the affine and inverse mapping processes are conducted multiple times. The authors need to provide a clear explanation of this aspect, as it may represent a significant deviation from their original proposal.

**Questions:**

Most of my primary concerns/questions are listed in the Weakness section.
Here, I listed my additional questions.

- Q1 (Method details): In lines 213 and 214, the author presents an additional normalization technique applied to the CLIP score results. I am interested in understanding the effectiveness of both the exponential transformation and the normalization process.

---

> ### Author Response · Authors · 2024-11-27
> **Authors' Response to Reviwer RYAn**
>
> - Reply to [W1]: Thank you for your constructive comments. We will add an ablation experiment to evaluate the sparsity of concept weights and demonstrate the effectiveness of the regularize. The takeaway of this experiment is that significant sparsity can be achieved without compromising the interpretability.
> - Reply to [W2]: Thank you for your comment. To clarify, the affine and inverse mapping process is applied to all the experiments across different datasets. The MLP mentioned in lines 355-360 is only part of the black-box model architecture we aim to explain. It does not change our proposed methodology. We will revise the text to make this clearer.
> - Reply to [Q1]: Thank you for your question. We have discussed the effectiveness of exponential transformation in Appendix A5,  particularly Fig.A.2. We discussed the effects of exponential transformation with power $t$ as 1 (i.e., no exponential transformation is applied), 3, and 5. The results show that $t = 5$ generally provides the best results. Regarding the normalization process, we believe it is intuitively necessary since both the original clip scores and the exponentially transformed clip scores lie in a very narrow range, making it hard to discriminate different concepts.

---

> > ### Comment · Reviewer_RYAn · 2024-12-02
> > **Response by Reviewer RYAN**
> >
> > Thank you for the authors' response. However, certain aspects of W1 remain unclear to me.
> > After carefully reviewing the responses and the comments from other reviewers, I'll maintain my score.

---

### Note · Authors · 2024-12-03

**Comment:**

We thank the reviewer's time and effort in reviewing this paper, and those constructive comments and suggestions. We have decided to withdraw it for more thorough improvements.

**Withdrawal Confirmation:**

I have read and agree with the venue's withdrawal policy on behalf of myself and my co-authors.